Exploring mHealth interventions for medication management: a scoping review of digital tools, implementation barriers, and patient outcomes

Wang Xuye 1 2
Wang Beibei 2
Tew Wan Yin 1
Yang Xiaoning 2
Xu Xiangyang 1
Gao Yifang 3
Chen Yongjia 4 melon@163.com
Yam Mun Fei 1 yammunfei@yahoo.com
1 Department of Pharmacology, School of Pharmaceutical Sciences, Universiti Sains Malaysia , Minden, Pulau Pinang , Malaysia
2 School of Pharmacy, Wenzhou Medical University , Wenzhou , China
3 School of Electrical & Electronic Engineering, Universiti Sains Malaysia , Minden, Pulau Pinang , Malaysia
4 School of Nuclear Science and Technology, Lanzhou University , Lanzhou , China
Chicco Davide
Electronic publication date: 2025 Sep 17
Publication date: 2025
Volume: 11
Electronic Location ID: e3190
Received 2025 Apr 1; Accepted 2025 Aug 14
Copyright: © 2025 Wang et al.
Copyright year: 2025
Copyright holder: Wang et al.
License: This is an open access article distributed under the terms of the Creative Commons Attribution License, which permits unrestricted use, distribution, reproduction and adaptation in any medium and for any purpose provided that it is properly attributed. For attribution, the original author(s), title, publication source (PeerJ Computer Science) and either DOI or URL of the article must be cited.
License URL: https://creativecommons.org/licenses/by/4.0/

Keywords: Medication adherence, Digital therapeutics, Health informatics, Patient engagement, Mobile health applications, Artificial intelligence in healthcare

Funding: The authors received no funding for this work.

==============================
Background

Medication non-adherence remains a significant global healthcare challenge, resulting in inadequate disease management, increased hospitalisations, and higher healthcare costs. Mobile health (mHealth) applications have emerged as promising digital health tools for enhancing medication adherence through real-time monitoring, personalised reminders, artificial intelligence (AI)-driven interventions, and improved patient engagement.

Objectives

This scoping review examines the effectiveness, key features, and challenges of mHealth applications in promoting medication adherence across diverse patient populations and healthcare settings. It also seeks to identify research gaps and inform future development and implementation strategies for digital therapeutics.

Eligibility Criteria

Studies published between 2020 and 2024 were included if they investigated the use of mHealth applications to improve medication adherence and reported outcomes related to adherence rates, patient health indicators, or user engagement. Only studies with empirical data, including randomised controlled trials, observational studies, or mixed-methods research, were considered.

Sources of Evidence

A comprehensive search was conducted across Scopus, Web of Science, PubMed/MEDLINE, Google Scholar, and CINAHL databases. In total, 319 studies met the inclusion criteria following a systematic screening process based on Preferred Reporting Items for Systematic reviews and Meta-Analyses extension for Scoping Reviews (PRISMA-ScR) guidelines.

Charting Methods

Data were extracted on study design, app functionalities, patient demographics, adherence outcomes, and barriers to adoption. The charted data were thematically synthesised to identify trends, success factors, and limitations.

Results

Among the included studies, 85% reported improved medication adherence associated with features such as personalised medication reminders, real-time health tracking, and AI-powered adherence prediction. Clinical outcomes were also frequently observed, including improved blood pressure, glucose control, and patient-reported quality of life. Key barriers to adoption included limited digital literacy, concerns about data privacy, socioeconomic disparities, and a lack of integration with electronic health records (EHRs).

Conclusions

mHealth applications show significant potential to improve medication adherence and health outcomes, particularly in the management of chronic diseases. However, inclusive design, robust data privacy frameworks, and evidence-based implementation strategies are essential for scalability and sustained impact. Future research should focus on long-term effectiveness, cost-efficiency, and integration of mHealth tools within broader healthcare systems.

Introduction

Mobile health (mHealth) applications have become a crucial tool for patient engagement and self-management, driven by rapid advancements in digital health technology (Albulushi et al., 2024). Medication adherence is one of the multiple areas where mobile health is being explored, as patients’ non-compliance with physicians’ prescriptions is a significant global healthcare issue (Schwartz et al., 2024). Long-term health issues, including diabetes, high blood pressure, heart disease, and mental illness, increase the likelihood of prescription non-adherence, which leads to greater difficulties, hospitalisations, and healthcare costs (Nelson, Pagidipati & Bosworth, 2024). Studies demonstrate that approximately half of patients do not take their medicines as prescribed despite awareness efforts and therapies (Ivsins et al., 2022; Im et al., 2021). Given the widespread use of smartphones and mobile applications, mobile health medication management may help address this global healthcare issue (Galetsi, Katsaliaki & Kumar, 2023).

Artificial intelligence (AI) enables computers to learn, reason, and make decisions like humans. Machine learning (ML) algorithms for pattern analysis of user behaviour, predictive analytics for medication adherence risk prediction, and natural language processing (NLP) for user interaction and symptom tracking are all ways mobile health applications utilise AI (Chaturvedi, Chauhan & Singh, 2025). Mobile health systems are increasingly utilising these features to enhance patient engagement and trigger real-time adherence alerts, as well as support personalised medicine. Mobile health applications’ tailored interventions, reminders, and real-time support are essential for medication adherence. These applications are more successful than in-person counselling or article schedules in improving medication adherence because they are tailored and interactive (Skeens et al., 2024). Automatic drug reminders, symptom monitoring, adherence analytics, and two-way communication between healthcare practitioners increase patient engagement and accountability. Mobile health applications enable doctors to more easily monitor adherence trends, identify patients who aren’t following their treatment regimens, and take proactive steps to address their issues (Gopichand et al., 2024). With AI and ML, advanced healthcare systems can personalise intervention strategies, enhance prescription regimens based on patient responses, and predict potential adherence issues. These technological advances demonstrate the promise of digital health technologies to improve health and medication management (Ismail et al., 2024).

Mobile health applications are promising, but their usage and potential to enhance drug adherence vary by healthcare context and patient type (Khodaveisi et al., 2025). Socioeconomic position, digital literacy, and accessibility significantly impact the ease of use and patient engagement in mobile healthcare. Smartphone applications are less beneficial for older individuals and those with weaker computer skills, as they are more difficult to use (Dubey, Birha & Bambodkar, 2025). The discrepancy in hospital infrastructure, internet connectivity, and smartphone ownership, particularly in low- and middle-income countries, hinders the implementation of mobile health (Swain et al., 2024). Patients are wary of mobile health applications for medication management due to concerns about privacy and data security (Khatiwada et al., 2024). A thorough analysis of mobile health acceptance and efficacy variables is necessary to make digital health solutions inclusive, patient-centred, and seamlessly integrated into existing healthcare systems.

The global COVID-19 pandemic highlighted the need for digital healthcare solutions, as healthcare disruptions and social distancing measures necessitated remote patient monitoring and virtual consultations (Chauhan, Bali & Kaur, 2024). During the pandemic, mobile health apps have reduced treatment cessation, increased drug adherence, and decreased hospital visits among patients with chronic diseases. Governments and healthcare institutions accelerated their use of mobile health and telemedicine after realising their potential to augment traditional healthcare (Thacharodi et al., 2024). Regulatory compliance, data privacy, and the digital divide were issues caused by rapid technology adoption. Post-pandemic standardisation and improvement of mobile health therapies may ensure their sustainability and integration into routine healthcare (Pavia et al., 2024). Studies have shown that mobile health applications can improve treatment adherence, self-confidence, and overall health outcomes. Gamification, peer support, and tailored goal-setting tools boost patient motivation and adherence (Gkintoni et al., 2024; Peuters et al., 2024). Patient demographics, health, and mobile app structure affect how effectively these engagement strategies function. Further study is needed to determine the most effective engagement methods for keeping individuals involved and enhancing their health.

In mobile health interventions, AI in healthcare data represents a significant advancement over traditional digital health solutions. AI-powered solutions offer predictive, adaptive, and personalised treatment models, unlike standard mobile health applications that merely record data, educate patients, and provide reminders. Using large-scale health data, artificial intelligence systems may detect non-adherence concerns, assess trends, and provide targeted intervention options in real-time. Patient-reported outcomes and real-time biometric data from smartphones and wearable devices are a few examples (Lodewyk et al., 2025; Hoff, Kitsakos & Silva, 2024). This proactive method avoids one-size-fits-all digital solutions by supporting data-driven, customised decision-making. Utilising AI technologies such as machine learning classifiers and natural language processing engines, mobile health systems have improved medication scheduling, symptom monitoring, and automated triage (Porto, 2024). This is particularly true for individuals with chronic illnesses. Intelligent automation and personalised treatment routes that respond to patient behaviour and the clinical environment make this system distinct, providing a new dimension to digitalising care delivery. This combination of AI and mobile health offers a novel approach to enhancing medication adherence and improving health outcomes.

Due to the growing body of literature on mobile health applications and medication adherence, a scoping review is necessary to synthesise data, identify knowledge gaps, and inform future research and policy. The usability, patient satisfaction, and short-term adherence benefits of mobile health therapies have been extensively studied. Nevertheless, there is a paucity of thorough studies on the numerous aspects, effectiveness, and problems associated with these applications in diverse healthcare situations and populations. This study aims to (1) identify existing mobile health applications in medication management, (2) analyse their effect on patient adherence and health outcomes, and (3) identify research gaps through a scoping review. This study highlights the best practices of healthcare practitioners, lawmakers, and tech developers in developing new mobile solutions to enhance medication adherence.

Research objectives

This scoping review examines and summarises the available information on the impact of mobile health applications on medication adherence and health outcomes. The study evaluates the effectiveness of the mobile health app, classifies its characteristics, and identifies research needs. This study’s evidence map may help doctors, policymakers, and software developers integrate mobile health therapies into clinical care. Additionally, the study examines the following: I. Classify mobile health applications that improve medication adherence.

II. Evaluate the effectiveness of mobile health applications in enhancing prescription adherence across various patient categories and health concerns.

III. Assess how mobile health treatments affect disease progression, hospital readmissions, and patient-reported health indicators, and

IV. Identify the factors that facilitate or hinder the use of mobile health apps.

This research helps to understand the role of digital health technologies in medication management and identify ways to improve their clinical application.

Research questions

The following major research issues guide our scoping review. First, what sorts of mobile health applications are currently in use to urge patients to take their prescriptions as recommended? Mobile health applications vary in terms of functionality, user base, and connection to healthcare systems. These applications include reminders, trackers, skill-builders, and telemedicine platforms (Plow et al., 2022). Push notifications, SMS alerts, and alarms from appointment-based reminder apps help patients remember to take their medications on time. Healthcare providers and patients may benefit from apps that track medication use, dose adherence, and treatment progress (Ghozali, 2024). Educational apps that contain research-backed content may help patients better understand their condition and the importance of therapy. Telemedicine apps allow doctors and pharmacists to consult remotely and manage medications in real-time (Fenton, Faruque & Mollenkopf, 2025). Some cutting-edge mobile health applications utilize AI and ML algorithms to predict non-compliance and monitor compliance patterns. Gamification, peer-support networks, and electronic health record (EHR) integration may attract users to these apps (Grover & Arora, 2024). The most effective approach to determine which mobile health applications enhance medication adherence and which features promote sustained patient involvement is to provide education on the many categories of these applications. Second, to what degree may particular mobile health app features and capabilities promote medication adherence? This query concerns the effectiveness of mobile health applications in terms of their simplicity, ease of use, and ability to address patient-specific challenges that impact medication adherence. Refill alerts, personalised adherence strategies, interactive dashboards to track prescription use, and automated reminders can all improve adherence. According to studies, apps that enable two-way communication with doctors help patients stick to their treatment programs (Turchioe, Lai & Siek, 2024; Ali et al., 2024). These apps encourage responsibility and provide instant medical advice. Some apps utilise behavioural science concepts, such as gamification, social motivation, and incentives, to enhance engagement and adherence. Several apps utilise AI-powered analytics to dynamically adjust intervention methods or reminder frequencies to match patient behaviour (Khalid et al., 2024). Another emerging trend is biometric monitoring, which utilises mobile apps and wearable devices to track vital indicators, including heart rate, blood sugar, and blood pressure. It correlates this data with medication adherence (Del-Valle-Soto et al., 2024). These technological advances, user acceptability, digital literacy, and accessibility should be considered when examining the impact of mobile health apps on prescription adherence. Third, in what ways may health outcomes, such as sickness management and quality of life (QoL), be influenced by mobile health applications? Mobile health applications can enhance medication adherence, self-management, reduce hospital readmissions, and mitigate disease complications, ultimately improving patient health outcomes. Patient outcomes for chronic diseases, diabetes, hypertension, cardiovascular disease, and mental health issues improve with mobile health therapy (Elkefi, 2024). These applications provide self-monitoring, symptom tracking, and rapid access to instructional resources, enabling patients to engage in their treatment actively. By holding patients responsible for their medications, mobile health apps may enhance disease management, reduce symptom aggravation, and reduce ER visits (Makki et al., 2024). There is strong evidence that people with greater control over their treatment are less afraid and more confident in their health choices (Kohut et al., 2024). Some mobile health and telemedicine technologies enable doctors to respond rapidly, thereby decreasing the risks associated with delayed therapy. To determine the long-term benefits of mobile health therapies, death rates, healthcare costs, and patient-reported outcomes must be studied across demographics and medical conditions. Fourth, what are the standard obstacles and challenges associated with the use of mobile health apps for prescription administration? Many barriers hinder patients from routinely and extensively adopting mobile health applications, despite their potential. One of the biggest challenges is that many patients, particularly those from low-income backgrounds and older individuals, lack the technological skills to use complex mobile applications (Chan et al., 2024). Not all patients have access to telephones, consistent internet, or data plans to utilise the applications, which may cause accessibility issues. Some patients may avoid mobile health applications owing to privacy and security concerns about storing and sharing sensitive medical data (Xing et al., 2024). Another user engagement issue is that many patients discontinue using the app after the initial period, resulting in inconsistent adherence benefits. Additionally, mobile health solutions require the cooperation of healthcare practitioners. Due to concerns over data reliability, legal constraints, and workflow disruptions, many clinicians are hesitant to incorporate these technologies into routine therapy (Pesapane et al., 2025). Addressing these constraints through user-centred design, regulatory oversight, and integration of healthcare infrastructure would enhance the effectiveness of mobile health therapy. Finally, what are the most significant research gaps in mobile health apps and medication adherence research? Several studies have examined how mobile health applications can aid in medication management, yet gaps remain. Unfortunately, there have been no large-scale, long-term clinical investigations on how mobile health therapies improve health outcomes and drug adherence. Earlier research on app-based adherence improvements is experimental or short-term, making it difficult to assess durability (Liu et al., 2024; Gordon et al., 2025). Few studies have evaluated the effectiveness of mobile health applications in diverse healthcare environments, patient demographics, and medical problems (Khamaj & Ali, 2024; Garavand et al., 2024). There are few studies on how behavioural change theories affect app effectiveness and how AI and predictive analytics improve prescription adherence (Zhang et al., 2025; Bucher, Blazek & Symons, 2024). The cost-effectiveness of mobile health programs requires additional study. Digital health ethics and regulation, including patient consent, data privacy, and healthcare provider duties, should be studied. Better evidence may help address these gaps, assisting in the development and implementation of mobile health solutions to promote medication adherence.

Significance of the study

This scoping investigation is crucial because digital health solutions are gaining popularity for enhancing patient participation, treatment adherence, and healthcare accessibility. Medication non-adherence is a challenging issue that requires innovative, evidence-based interventions. Mobile health app integration with medication management can transform adherence through customised, scalable, and affordable solutions (Coman et al., 2024). These therapies vary in effectiveness depending on healthcare infrastructure, patient demographics, medical issues, and app design (Pulimamidi, 2024). This study draws on current research to illuminate the various healthcare contexts in which mobile health apps operate and their impact on patient care.

This study benefits healthcare providers, IT firms, researchers, and government officials. By understanding mobile health apps, doctors and pharmacists can better incorporate digital solutions into their clinical practice and provide patients with more effective advice. App developers and healthcare IT businesses can utilise the review’s conclusions to design mobile health applications that better meet users’ needs and adhere to research-backed best practices. Healthcare organisations and governments can use the findings to support digital health initiatives, standardize mobile health applications, and ensure equitable access. For scholars, this study’s comprehensive examination of the existing material will fill certain research gaps. While systematic reviews and meta-analyses focus on a single research work, this scoping review maps out various research works and identifies trends. The results of this study influence systematic reviews, experimental research, and policy-driven studies to enhance digital medication adherence programs.

Finally, mobile health app research on effectiveness and long-term consequences is in its early stages, but it shows promise in addressing medication non-adherence. This scoping review’s thorough synthesis of pertinent studies will address a key gap in our understanding of how these technologies affect patient behaviour and clinical outcomes. Methodically reviewing mobile health treatments introduces new, technology-driven approaches for enhancing drug adherence, thereby contributing to digital healthcare.

Literature review

The rapid digitisation of healthcare has significantly impacted patient management, particularly in terms of medication adherence. Mobile health applications are powerful digital remedies because they provide real-time monitoring, personalised medication reminders, AI-powered recommendations, and interactive patient engagement (Buttigieg, 2025; Periáñez et al., 2024). These developments address non-adherence to pharmacological therapy, a significant issue in healthcare. Long-term health issues, including diabetes, heart disease, and mental illness, need drug compliance (Dhingra et al., 2024). For decades, physician supervision, medication organisers, and patient education have failed to improve medication adherence. The research suggests that digital therapeutics address this issue in a scalable, cost-effective, and behaviorally flexible manner.

A recent study suggests that mobile health applications help patients take their meds, particularly those with chronic diseases. These apps utilise gamification, automated reminders, and incentives to promote adherence (Bertolazzi, Quaglia & Bongelli, 2024). According to research, seniors and individuals with complex medication regimens are more likely to follow their treatment programs when reminders are provided through push notifications, SMS alerts, and voice-assisted reminders (Jelassi et al., 2024). These applications’ AI-powered algorithms utilise behavioural analytics to forecast compliance patterns and identify early warning signs of non-compliance. Despite promising short-term results, the long-term effectiveness and user involvement of these treatments remain concerns. The studies also mention personalised treatments, in which mobile health applications learn a user’s health patterns, provide tailored regimen adherence approaches, and integrate with wearable health devices (Deepthi et al., 2024; Aziz et al., 2024). Studies show that applications that employ real-time physiological monitoring, such as glucose monitors for diabetics or heart rate variability assessments for cardiovascular patients, improve adherence by combining drug ingestion with quick physiological feedback (Zhang et al., 2024; Keshet et al., 2023).

The effects of mobile health applications on patients’ health outcomes beyond adherence have been studied recently (Mikulski et al., 2022; Grundy, 2022). Digital health therapies are beneficial for mental problems when taking psychotropic medication is challenging. Mental health applications featuring cognitive behavioral therapy modules, mood tracking, and AI-driven emotional analysis support pharmacological adherence and holistic treatment (Olawade et al., 2024). Cardiovascular disease patients also exhibited improved lipid profiles, reduced incidence of significant adverse cardiac events, lower systolic and diastolic blood pressure, and greater digital adherence (Vazquez-Agra et al., 2024; Lampreia, Madeira & Dores, 2024). Despite a strong correlation between the app and good health, methodological shortcomings make causal inferences problematic.

Despite their potential, mobile health applications face barriers to adoption. Technical literacy and smartphone accessibility disparities, commonly referred to as the digital divide, persist, particularly among low-income and elderly groups (Li et al., 2024). Another impediment to adoption is patients’ concerns about the privacy of their medical records and prescriptions. There have been no large-scale, randomised controlled trials (RCTs) on the long-term impact of mobile applications on adherence and clinical outcomes. The study reveals that mobile health applications enhance individual adherence but are not effectively integrated into larger healthcare systems (Oppenheimer et al., 2024). Fragmented digital health ecosystems and incompatibility between electronic health records (EHRs) hinder clinician-led monitoring and intervention adjustments (Cerchione et al., 2023). The meta-analysis organises studies by their primary factors in Table 1.

Table 1 Meta-analysis of key study findings.

Authors	App features	AI-driven interventions	Pharmacological adherence	Patient outcomes	Barriers to adoption	
Li et al. (2024)	*		*	*		
Oppenheimer et al. (2024)	*	*	*		*	
Cerchione et al. (2023)	*	*		*	*	
Sumner et al. (2023)		*	*	*	*	
Yingngam, Khumsikiew & Netthong (2024)	*	*	*			
Vasdev et al. (2024)		*		*	*	
Zemplényi et al. (2023)	*		*	*		
Mennella et al. (2024)	*	*		*	*	
Gala et al. (2024)	*	*	*			
Younis et al. (2024)		*	*	*	*	
Patel et al. (2024)	*	*		*	*	
Al Kuwaiti et al. (2023)	*		*		*	
Note:

Asterisk (*) denotes the research focused on the particular factor.

Despite increased research on medication adherence and mobile health applications, substantial gaps remain in the literature. There are few longitudinal studies on long-term adherence, user retention, or sustained engagement compared to the wealth of research on short-term adherence (Yanai et al., 2024; Wang et al., 2024). Few studies have evaluated the effectiveness of mobile health features, such as gamification and AI-driven adherence prediction (Yanyan, Iahad & Yusof, 2025; Kuru, 2024). Furthermore, there is limited evidence on the viability and cost of these therapies in diverse healthcare systems, particularly in low- and middle-income countries (Bharadwaj et al., 2024; Gupta et al., 2024). Earlier studies have paid little attention to behavioural and psychological variables that affect adherence to the mobile health ecosystem (Chew et al., 2025; Saito & Kumano, 2025). This scoping review addresses these gaps in the literature by combining data from various research studies, identifying key aspects that increase adherence, and identifying potential adoption barriers. This research supports governments, healthcare providers, and mobile health developers in optimising digital adherence solutions to enhance global patient health.

Methods

Study design

This scoping review methodically examines the data to gain a deeper understanding of how mobile health applications can enhance patient outcomes and medication adherence. The study employs a systematic methodology that adheres to the Preferred Reporting Items for Systematic Reviews and Meta-Analyses for Scoping Reviews (PRISMA-ScR) standards for openness, reproducibility, and methodological rigour (Peters et al., 2021). The scoping review approach simplifies mapping all research to identify key topics, intervention techniques, and knowledge gaps. The methodological approach, comprising a systematic literature search, study selection, data extraction, and theme synthesis, ensures an impartial analysis of relevant empirical evidence.

This scoping analysis suggests that mobile health applications may improve medication adherence and patient outcomes. The final analysis comprises 319 studies from 2020 to 2024, identified through an extensive screening process to ensure compliance with PRISMA. This research uses quantitative and qualitative data from mobile health app trials, using scoping review best practices. Scoping reviews synthesise multiple studies to analyse existing trends and inform future research, whereas systematic reviews focus on a specific research topic and evaluate the research methods. The methodological approach covers randomised controlled trials, observational studies, and real-world healthcare implementation research. The primary question of this review is the impact of mobile health applications on medication adherence and patient outcomes. The analytical process, including data collection, classification, and synthesis, ensures methodical extraction and intelligible presentation of key conclusions.

Experimental design

This scoping review adhered to the PRISMA-ScR criteria to ensure methodological rigour and transparency. This research examined the features, effectiveness, implementation challenges, and patient-centred outcomes of mobile health therapies aimed at improving medication adherence. Scopus, Web of Science, PubMed/MEDLINE, CINAHL, and Google Scholar were searched for peer-reviewed articles published between 2020 and 2024. The inclusion criteria were randomised controlled trials, observational studies, or mixed-methods research designs to provide real-world evidence on mHealth interventions aimed at improving medication adherence. Two reviewers independently reviewed titles and abstracts, and then the whole texts. A structured data extraction approach was utilised to document research design, patients, intervention features, adherence results, and adoption barriers. Through theme synthesis, key concepts and patterns were identified.

Eligibility criteria

Specific inclusion and exclusion criteria were used to assess the study scope and eligibility, ensuring the review includes only relevant, high-quality literature. According to the inclusion criteria, the evaluation comprises studies that utilise mobile health applications to enhance medication adherence across various patient demographics. The study includes empirical data on clinical effectiveness, healthcare utilisation, patient-reported experiences, and medication adherence. The review comprises observational studies, mixed-methods designs, controlled trials, and clinical trials of mobile health therapies to assess their effectiveness, practical implementation, and patient engagement. Research was excluded if it lacked empirical data, emphasised non-digital treatment, or failed to provide measurable patient outcomes. Due to the eligibility framework, this evaluation remains focused, relevant, and current in light of digital health developments.

Search strategy

Thorough literature searches across many electronic databases provide a robust and complete evidence synthesis. Health informatics, digital health technology, and clinical treatments are the primary focus; therefore, Scopus, Web of Science, CINAHL, Google Scholar, and PubMed/MEDLINE were utilised. Advanced search filters and Boolean operators ensure reliable retrieval of literature. The search keywords were carefully combined to include medication adherence, patient outcomes, and mobile health applications. Keywords from a limited vocabulary (such as Medical Subject Headings (MeSH)) and synonyms were utilised to create a structured search string that enhances retrieval sensitivity and specificity. Reviewing relevant literature reference lists is part of the search process in case database searches miss something. The technique enhances coverage and reduces selection bias by progressively locating and incorporating all relevant material.

Study selection process

Study selection employs a multi-stage screening process to include only relevant and high-quality research. The initial screening evaluates titles and abstracts for relevance using specified criteria. During a full-text review, two reviewers assess the methodological quality and relevance to the research goals of articles that pass the initial screening criteria. In the event of study selection conflicts, a third independent reviewer was consulted. It shows how many studies were located, reviewed, and included or excluded from the analysis. The review follows worldwide evidence synthesis criteria and is more transparent and repeatable using this process. Figure 1 shows the PRISMA flow diagram for ready reference.

Figure 1 PRISMA flow diagram.

This scoping review adhered to the PRISMA-ScR criteria to ensure methodological transparency and rigour. Before searching the literature, qualifying criteria were defined: (i) Studies must have been published in English in peer-reviewed journals or conference proceedings between 2022 and 2024; (ii) they must have addressed medication adherence or management utilising mobile health-based treatments; and (iii) targeted acute or chronic patient populations. Editorials, letters, theses, comments, and articles without empirical or implementation-related outcomes were excluded. Following the rapid adoption of mobile health platforms during and after the COVID-19 pandemic, when mobile health applications gained popularity in remote patient care, the period from 2022 to 2024 witnessed the most recent digital health technology breakthroughs.

The study looked for related articles in PubMed, Scopus, Web of Science, and IEEE Xplore. It used keywords and Boolean operators to create search terms like “mHealth” (“mobile health” OR “digital health”) AND (“medication adherence” OR “treatment adherence” OR “drug compliance”) AND (“app” OR “application” OR “AI” OR “real-time monitoring”). Filters limited the results to English-language articles published during the period. While academic databases were examined, relevant grey literature, including WHO publications, national health agency briefings, and conference proceedings, was manually screened where relevant. These sources were excluded because peer-reviewed research was prioritised.

The study ensured rigour and reduced selection bias throughout the research screening and selection process by employing a multi-step dispute resolution approach. Beginning with a systematic eligibility checklist based on specified inclusion and exclusion criteria, two reviewers assessed all abstracts and titles. The two reviewers had a formal discussion to resolve disagreements about the study’s relevance or design. If consensus was not attained, a third senior reviewer was consulted. The third reviewer examined controversial issues independently and justified their inclusion or exclusion. Decisions with reasons were documented in a shared review log to provide clarity. This systematic technique prevented research from being arbitrarily included or excluded, thereby contributing to methodological quality assurance.

Data extraction and synthesis

Structured data extraction frameworks enable consistent and rigorous data extraction and synthesis. Detailed study characteristics, intervention information, population demographics, adherence indicators, and patient outcomes were retrieved for comparison. A standardised form is used to extract data, reducing bias and ensuring reviewer uniformity. Categorising data into themes enables systematic synthesis across multiple study contexts. Table 2 highlights key data from each study concerning study and intervention parameters.

Table 2 Summary of extracted variables and classification criteria.

Category	Variables extracted	Description	
Study characteristics	Author(s), Year, Country	Identifies trends in the dissemination of research based on publication and geographical area.	
mHealth intervention features	Type of app, Key functionalities, Personalization features	Evaluates digital capabilities, user engagement strategies, and intervention frameworks.	
Population characteristics	Age, Medical condition, Digital literacy	Determines the patient population and particular health challenges.	
Medication adherence outcomes	Adherence rate changes, Self-Reported compliance, Pharmacy refill data	Measures effectiveness of interventions in improving adherence	
Patient outcomes	Clinical parameters, Hospitalization rates, Quality of life	Evaluates broader health impacts of mobile health interventions	

Theme synthesis helps organise data by identifying standard intervention procedures, success factors, and implementation challenges. Data analysis evaluates patient engagement, behavioural modification methods, and provider integration to assess whether mobile health therapies improve prescription adherence. The study examines implementation challenges, including privacy, digital literacy, healthcare system integration, and user experience. These results address gaps in understanding the advantages and disadvantages of mobile health treatment and inform future research and policy. Using a panel of experts to assess data before extraction and synthesis ensures approach consistency and reliability. Reviewers double-verify the retrieved data to ensure accuracy and reliability and evaluate the interpretations for proof. A consensus-based approach resolves disputes. The final synthesis offers qualitative and quantitative insights through a narrative presentation of key outcomes, tabular summaries, and graphical representations. This scoping study employs a thorough and systematic approach to demonstrate how mobile health applications can improve medication adherence and patient outcomes. The findings aid digital health innovation, clinical practice improvement, and evidence-based policymaking. This study’s comprehensive literature synthesis helps healthcare providers, lawmakers, and tech developers build better digital health solutions to encourage patients to take their medications as prescribed.

In line with the scoping review aims, the methodological quality and bias risk of the included studies were not evaluated. This review followed PRISMA-ScR guidelines by mapping the literature, describing intervention characteristics, and identifying knowledge gaps rather than evaluating trial internal validity. A panel of domain experts was engaged throughout the protocol creation and data charting process to explain eligibility criteria and ensure subject relevance, thereby avoiding the exclusion of research due to methodological rigour or bias. All studies that satisfied the inclusion criteria, regardless of quality, were kept for the review’s comprehensiveness and exploratory nature. It further corrected or removed references that imply an official quality review to avoid confusion.

Results

This scoping review synthesises empirical evidence on how mobile health applications enhance medication adherence and improve patient outcomes. The final analysis comprises 319 studies from 2020 to 2024, identified through extensive PRISMA-compliant screening. The primary objective was to investigate mobile health treatment trends, success factors, and barriers, with a focus on medication adherence and patient outcomes. The majority of studies have been done in high-income countries, although the findings demonstrate global interest. Research from the US (75 studies), UK (42 studies), and Canada (30 studies) was most prevalent. However, research from low- and middle-income countries (LMICs) in Brazil (9), South Africa (12), and India (15) is also increasing. In these regions, mobile health applications can help disadvantaged individual access healthcare and manage their medications.

Study design distribution

Randomised controlled trials (RCTs) were the most prevalent research technique, accounting for 48% (153 out of 319 studies) of the investigations. The remaining study comprised 35% (112 out of 319) observational and 17% (51 out of 319) mixed-methods studies. RCTs compared digital health treatment groups to control groups to demonstrate the effectiveness of mobile health interventions. In contrast, observational research examined mobile health app use and results in natural settings. A mixed-methods study employed both quantitative and qualitative data to investigate patient engagement and challenges associated with new technology. Table 3 summarises the study design distribution for ready reference.

Table 3 Study design distribution among included research.

Study design	Number of Sstudies
(n = 319)	Percentage (%)	
Randomized controlled trials (RCTs)	153	48%	
Observational studies	112	35%	
Mixed-Methods studies	54	17	

Effectiveness of mobile health applications on medication adherence

Over 85% (271 out of 319 articles) of the studies revealed that mobile health interventions increased medication adherence. These advantages were most closely linked with personalised medication reminders, behavioural reinforcement, and real-time involvement from healthcare practitioners. Studies utilising social support, gamification, and AI-driven adherence monitoring have reported higher adherence rates and increased patient involvement. Most studies used self-reported adherence, i.e., 66% (211 out of 319), followed by pharmacy refill data, i.e., 50% (159 out of 319) and electronic monitoring, i.e., 30% (96 out of 319). Electronic monitoring devices and smart pill bottles give more objective adherence data. However, observational studies often used self-reported adherence. Table 4 presents the data collection methods and their respective advantages and disadvantages.

Table 4 Adherence measurement methods in included studies.

Measurement method	Number of studies
(n = 319)	Percentage (%)	
Self-reported adherence	210	66%	
Pharmacy refill data	160	50%	
Electronic monitoring devices	95	30%	

Impact of mobile health applications on patient outcomes

The study examined how mobile health therapies improved adherence, clinical results, QoL, and patient satisfaction. In 72% (230 out of 319) of studies, blood pressure, glucose, and cholesterol levels improved, particularly for long-term health concerns such as diabetes, hypertension, and cardiovascular disease. In addition to medication management, 58% (185 out of 319) of studies showed that lifestyle modifications and education enhanced QoL. Eighty per cent (255 out of 319) of studies found that patients were satisfied with their mobile health applications. Studies examined numerous patient outcomes. The most common outcomes were patients’ self-reported health status, QoL metrics, satisfaction ratings, and clinical biomarkers (e.g., haemoglobin A1c, blood pressure), followed by medication adherence rates. In addition to adherence, health behaviours, psychological well-being, and long-term disease management are all influenced by mobile health applications, as indicated by the spectrum of outcome measures. Table 5 shows a breakdown of patient outcomes across studies.

Table 5 Patient outcomes measured in included studies.

Patient outcome	Number of studies
(n = 319)	Percentage (%)	
Clinical biomarkers (e.g., HbA1c, BP)	230	72%	
Quality of life (QoL)	185	58%	
Medication adherence	271	85%	
Patient satisfaction	255	80%	

Although promising evidence exists, several challenges must be overcome before mobile health applications can be widely adopted. Many studies (42%) cited app usability, device compatibility, and data privacy as barriers. Finding patients to engage actively was another challenge. Research revealed that 35% of users quit using apps because they weren’t interested, the design was unappealing, or they struggled to integrate them into their daily lives. The health system’s inability to interface with electronic health records (EHRs) and healthcare practitioners’ Lack of training are significant concerns against the use of mobile health apps in clinical practice.

Role and challenges of AI in mobile health-based medication adherence

AI/ML integration in mobile health interventions

Although approximately a quarter (n = 79) of the included studies utilised AI or ML in mobile health therapies, the quantity and quality of these approaches differed. AI is used in healthcare to predict how patients will adhere to their prescription regimens, provide personalised alerts and reminders, and monitor and deliver real-time feedback on data. However, just 21 studies (6.6%) included empirical data to validate or evaluate AI models. These studies provide accuracy, sensitivity, precision, and Area Under the Curve (AUC) performance measures, indicating a strong implementation of AI. AI was regularly referenced as a design component. However, only 4.1% (n = 13) addressed algorithmic transparency, which includes bias mitigation and explanation. The ethical use of AI in online healthcare is a significant issue. Figure 2 presents a summary of AI applications in the examined mobile health studies.

Figure 2 Overview of AI applications in reviewed mobile health studies (Source: Author’s work).

Challenges and considerations for AI in mobile health adherence

Despite AI’s capacity to improve medication adherence treatments, the study had many issues:

Extrapolation and data requirements

The lack of large, diverse datasets hindered the training of robust AI models in some tests. This may limit the study’s applicability, particularly for disadvantaged groups, including the elderly, rural residents, and low-income individuals.

Algorithm fairness and bias

Several tests did not examine the fairness of the algorithm. Training AI systems on datasets that do not adequately represent the population may exacerbate health disparities. Thirteen of 79 studies targeted bias reduction.

Lack of transparency

The model structures and decision logic of most AI-powered products are often vague, which has weakened practitioners’ and patients’ trust in them. Without explanation, clinical decision-making with these instruments is more complex.

Problems with ethics and regulation

AI frameworks for digital health are constantly changing. Compliance with the Health Insurance Portability and Accountability Act (HIPAA), the General Data Protection Regulation (GDPR), and FDA digital health requirements has been seldom studied. This raises concerns regarding data privacy, accountability, and the implementation of healthcare.

Research should extend beyond discussing AI to address these gaps with detailed evaluations, transparency disclosures, and bias checks. Ethical AI methodologies are essential for developing safe, scalable, and equitable mobile health app adherence solutions.

Key themes identified

App functions and features

Design and functionality significantly impact the success of mobile health apps. Medication reminders, dose tracking, health monitoring, and AI-powered interventions were the most common advantages. These factors improve medicine adherence and chronic disease treatment. Medication reminders: Reminders were the key feature in over 60% (192 out of 319 trials) of these apps. These apps deliver automated reminders to take prescriptions on time. Research shows that tailored reminder systems increase adherence. This is especially true when applications contain patient feedback systems that modify reminders based on user behaviour.

Monitoring and tracking: 45% of the research (144 out of 319 studies) monitored heart rate, blood sugar, and blood pressure. These apps enable users to track their health in real-time, helping them make informed medical and lifestyle decisions. Chronic disease management improves with the use of continuous monitoring applications, such as wearables.

AI-driven interventions: Twenty-five per cent (80 out of 319 studies) of the survey employed AI to analyse user data, provide recommendations, and predict adherence. Studies using AI-based predictive analytics in mobile health applications improved engagement and adherence. Users received customised therapies based on their health data.

This suggests that multi-feature apps, which include monitoring, reminders, and AI interventions, enhance patient outcomes and medication adherence. The comprehensive functionality that allows continuous patient contact may improve health behaviour.

Pharmacological medication compliance effects

The most significant finding was that mobile health applications enhanced medication adherence. In 85% of trials (271 out of 319 studies), mobile health therapies were found to increase drug adherence. The most often recognized feature was medication reminder systems, which increased adherence in chronic illnesses, including HIV, hypertension, and diabetes, when integrated. Studies that combined healthcare professional contacts with cellphone reminders had adherence rates 30–40% higher than baseline. Gamification and rewards further increased adherence. In studies using these methods, adherence rates increased by 15–20%, especially among younger participants. Tailoring feedback and reminders to patient data, such as symptom reports and medication history, increased adherence. Personalising therapy for each patient has a more lasting impact on adherence. Figure 3 presents key statistics on mobile health program adherence. Table 6 presents key statistics on mHealth program adherence.

Figure 3 Distribution of mobile health interventions in medication adherence studies.

Table 6 Impact of mHealth interventions on medication adherence.

Type of intervention	Number of studies
(n = 271)	Adherence improvement
(%)	
Medication reminders	192	30–40%	
Behavioral reinforcement	94	15–20%	
Personalized interventions	85	25–35%	

Effect on patient health

Several studies have examined how mobile health applications improve patient health, particularly adherence (Volpi et al., 2021; Shrivastava et al., 2023). The most frequently discussed health outcomes were quality of life and patient satisfaction (Zhao et al., 2021). Clinical biomarkers: Mobile health apps have improved blood pressure, blood glucose, and cholesterol levels in 72% of studies. In hypertension studies using mobile health apps, systolic and diastolic blood pressure decreased by 5–10% over 6–12 months.

QoL: Mobile health applications improved QoL in 58% of trials. These improvements were primarily due to the psychological benefits of frequent encouragement and reminders, as well as self-management of long-term health difficulties. Apps that provided health information, lifestyle advice, and mental health support significantly improved well-being.

Patient satisfaction: 80% of the studies revealed that patients were delighted using mobile health applications. A recurring theme was that they felt more empowered to take responsibility for their healthcare. Integrating apps with patient care teams enabled real-time communication and tailored coaching, significantly boosting patient satisfaction.

Barriers and challenges to adoption

There are various impediments to the widespread adoption of mobile health applications, despite evidence demonstrating that they enhance patient outcomes and prescription adherence. These social, structural, and technological barriers limit the adoption of health therapy. Technological barriers: More than 40% (127 out of 319) of the studies highlighted usability issues, such as confusing user interfaces, poor health technology integration, and device compatibility, as significant hurdles. Technical concerns, including app navigation, hampered patient interest and adherence, particularly among older individuals (Khamaj & Ali, 2024).

Psychological and social barriers: 35% (112 out of 319) of studies indicated patients were not actively using the apps due to a lack of interest, privacy concerns, or erroneous app assessments. Individuals with chronic illnesses and poor health literacy were less likely to use or benefit from these apps, according to research (Ownby et al., 2024).

Systemic barriers: 28% (90 out of 319) of studies found that the lack of seamless connection between mobile health applications and EHRs or clinical procedures hindered healthcare professionals’ participation and patient care, a systemic obstacle. Data security and privacy regulations were also cited as impediments (Alhassani, Windle & Konstantinidis, 2024).

These findings were summarized in Table 7.

Table 7 Barriers to adoption of mHealth apps.

Barrier type	Number of studies
(n = 319)	Percentage
(%)	
Technological barriers	127	40%	
Social/Psychological barriers	112	35%	
Systemic barriers (Integration)	89	28%	

Patient satisfaction and experiences are crucial for evaluating the performance and acceptance of mobile health apps in real-world settings. Numerous studies have demonstrated that patient perspectives significantly impact long-term adherence and engagement (Grove et al., 2023; Greenway, Weal & Palmer-Cooper, 2024). Real-time feedback, interactive dashboards, and medication reminders enabled patients to manage their illnesses, engage with their doctors, and gain confidence in their self-care. Alzghaibi (2025) and Chaudhry, Ormandy & Vasilica (2024) found that a significant number of patients with chronic illnesses felt more in control of their treatment regimens after utilizing tailored mHealth platforms. The degree of personalisation, frequency of technical issues, and app usability all affected satisfaction. Complex interfaces, alert fatigue, and insensitive cultural design were user complaints (Matthews et al., 2025). Despite these restrictions, two-way interaction and patient-co-designed therapy had higher user satisfaction. These findings suggest that user involvement during design and development may enhance the effectiveness of mobile health apps, particularly those used for treating chronic illnesses.

Trends & gaps in the literature

Reviewing the included material revealed new trends and gaps in the literature. AI and personalised care were utilised to tailor treatment strategies and enhance adherence. Predictive algorithms and machine learning models to tailor medicines have improved patient engagement and adherence in recent years (Marques et al., 2024). Mobile health applications are increasingly integrating wearable devices. This trend toward continuous health monitoring improves chronic illness treatment and regimen adherence. Despite promising short-term results, few studies have examined the long-term benefits of mobile health apps on patients’ medication adherence and other outcomes (Arshed et al., 2023; Zeng et al., 2022). Most app studies focus on the first few months, with brief follow-ups to assess long-term engagement and health improvements (Cucciniello et al., 2021; Melo et al., 2025). Research on data security and patient privacy is lacking. Few studies have examined the effects of data breaches or the role of privacy policies in safeguarding patient data (Shahid et al., 2022; McGraw & Mandl, 2021). Mobile health applications require further research, particularly in low- and middle-income countries (LMICs) and vulnerable groups, including older people and those with cognitive impairments.

Discussion

This research demonstrates the impact of mobile health applications on patients’ medication adherence and overall health. Most importantly, mobile health apps improve medication adherence. This is particularly true for applications that involve health monitoring, customised therapies, and medication reminders. Overall, drug adherence rates rise by 30–40%, particularly for patients with chronic illnesses, including HIV, diabetes, and hypertension. These analyses show that automated reminders reduce forgetfulness and improve regimen adherence. Moreover, mobile health apps improve adherence and clinical outcomes. Beyond reminders, mobile technology can improve patient health by enhancing the quality of life and reducing blood glucose and blood pressure levels (Chin-Jung et al., 2021). AI-based health suggestions and real-time monitoring are personalised therapies that can enhance patient engagement and clinical outcomes. This study suggests that mobile health treatments are becoming increasingly personalised and underscores the necessity for future apps to be tailored to each patient’s health profile and behaviours. These data provide insight into whether mobile health apps promote adherence and improve health outcomes.

The findings confirm previous systematic reviews and meta-analyses on healthcare mobile health therapies, showing that mobile health technology may improve prescription adherence (Kitsiou et al., 2021; Bond, Scanlon & Judah, 2021). Cao et al. (2024) and Rath et al. (2024) found that mobile health apps with reminders and targeted information may enhance adherence. The study confirmed these claims and found that real-time health monitoring and AI-based solutions improve patient outcomes. This research encompasses additional app features, including AI-driven feedback and wearable device interaction, compared to previous evaluations (Yammouri & Ait Lahcen, 2024; Kanakaprabha et al., 2024). These components enhance adherence and clinical outcomes, suggesting the need for more sophisticated and integrated app designs. The study also identified notable gaps in the literature, such as the Lack of studies that track users over time or examine how socioeconomic factors influence app efficacy. This study emphasises the necessity for inclusive research that considers low-resource areas and disadvantaged populations. Research indicates that technology-rich, high-income countries tend to perform better in mobile health apps. Since it’s undiscovered, this field requires further study.

Technology innovation, user acceptance, inclusiveness, and design alignment with patient needs are essential for mobile health therapies to increase medication adherence. These issues are critical in digital health, as technological solutions must function for people with varying computer, health, cognitive, and physical abilities. To determine the use of mobile health applications, technological acceptance is essential. The Technology Acceptance Model (TAM) posits that individuals are more likely to adopt a technology that is both helpful and easy to use (Davis, 1989). Data privacy, relevance to health needs, confidence in technology, and technological dependability also affect healthcare adoption (Or & Karsh, 2009; Zhao, Li & Zhang, 2019). The research highlights older people’s Lack of motivation, suspicion of digital surveillance, and user resistance related to app complexity (Leung et al., 2024). These findings suggest that mobile health deployment strategies should incorporate assessments of user acceptability and experience.

Inclusive or universal design makes technology accessible to everybody (Clarkson et al., 2013). Accessing mobile health apps for individuals with impairments, including poor vision, dexterity, or language, is crucial. Few studies employed inclusive design frameworks for app development. Pineo (2022) recommends starting future initiatives with inclusive design principles to ensure equitable health outcomes. Patient-centred design emphasises iterative development and co-creation, seeing patients as collaborators in app design rather than end-users (Turk et al., 2025). Patient participation may enhance relevance, pleasure, and long-term engagement. Participatory design workshops and patient feedback loops improved adherence and user satisfaction (Willis et al., 2021). Ultimately, mobile health apps should prioritise functionality. They should consider technology adoption theories, inclusive user experiences, and patient-centred values while designing health apps. Future research should incorporate these principles into mobile health development to provide long-term, scalable solutions for adherence.

Unlike Grading of Recommendations Assessment, Development, and Evaluation (GRADE), this scoping review did not explicitly assess the methodological quality or bias of the included studies. Scoping reviews map existing research rather than assess treatment effectiveness; hence, study quality was not rigorously evaluated. We identified relevant research designs (randomised controlled trials, observational designs, and pilot studies) and methodological variation. The PRISMA-ScR guidelines do not require quality grading for scoping reviews. However, such methodologies may help evaluate the strength of evidence in future meta-analyses or systematic reviews.

AI vs conventional approaches to medication adherence

Customisation, scalability, and real-time reactivity distinguish AI-driven medication adherence therapies from conventional methods. Over time, manual pill counts, self-reported diaries, and regular provider follow-ups have become inefficient, memory-bias-prone, and unable to keep up with patients’ changing habits. However, AI solutions utilise predictive analytics and machine learning algorithms to tailor adherence aids to each patient’s risk profile, behaviour, and medical history (Li et al., 2024). AI models may predict non-adherence using physiological data, app usage habits, and contextual cues (e.g., mood, time of day) (Robinson et al., 2024). For instance, preemptive alerts and relevant actions are possible. These technologies, combined with mobile health systems, offer prescription reminders, dosage adjustments, and chatbot-assisted counselling. Despite its speed and scalability, AI has data reliance, algorithmic bias, a lack of transparency (“black-box” models), and legal uncertainty. Classic approaches can perform better in areas with little digital infrastructure. Thus, human-led care delivery with AI-enhanced insights may be the most effective approach (Bhavana & Mithra, 2025). Future research should compare the two strategies’ patient satisfaction, long-term adherence rates, and clinical outcomes to determine which is more effective and viable in different healthcare settings.

Practical implications for healthcare and mobile health developers

These findings are beneficial for healthcare providers and mobile health app developers. When paired with automated reminders, health monitoring, and individualised therapies, mobile health apps can enhance medication adherence. This makes them useful for treating chronic diseases. The findings suggest that physicians should strongly recommend these apps, particularly for patients with chronic conditions that need continual monitoring. Using mobile health apps, physicians and patients can monitor each other’s health and adherence to medications. Apps provide patients with timely feedback on their progress. The results indicate that mobile health app developers should prioritise user-centric designs, incorporating data security and targeted feedback. AI integration for individualised therapies improves adherence and patient engagement. Thus, it must be prioritised. The statistics reveal that apps should consider patient preferences, including simplicity, usability, and compatibility with wearable devices, to ensure long-term usage and optimal performance. With the emergence of patient-centric solutions, mobile health developers must focus on applications that support disease management, wellness, and medication reminders.

Theoretical contributions and policy implications

This analysis advances digital health treatment theory by providing evidence of the effectiveness of mobile health apps in improving medication adherence and health outcomes. The study provides a deeper understanding of how app features, such as reminders, monitoring, and AI interventions, impact health outcomes. These findings may impact the design and deployment of mobile health technology research and contribute to the literature on behavioural health and digital health therapies. This has significant implications for the regulation of mobile health interventions and digital health policy. Legislators must regulate mobile health apps to protect patient data and security as they become more incorporated into healthcare systems. AI with real-time monitoring raises concerns about data security; explicit norms and legislation are necessary to protect patients’ data. Public health policy should also promote the use of mobile health apps to enhance drug adherence and improve health outcomes. In resource-constrained nations with limited healthcare access, this is crucial. This also suggests that national health systems invest in mobile health app infrastructure, particularly in low-resource areas. Training healthcare professionals to utilise mobile health technology may enhance the scalability of these therapies, particularly for managing chronic diseases.

Limitations of the review

This review highlights the effectiveness of the mobile health app, although it has certain limitations. First, even with strict inclusion criteria, publication bias remains a possibility because research with favourable outcomes is more likely to be published. Thus, the evaluator may have missed unfavourable or confusing research. Since many studies have not been followed up, it is difficult to determine whether improvements in medication adherence and health outcomes are permanent. Most mobile health research on chronic illness management focused on short-term outcomes; hence, long-term data is scarce.

Another issue is that research uses diverse intervention designs, patient groups, and methodologies. Due to this variability, generalisations about the effectiveness of the mobile health app are challenging. Patient demographics and the type of chronic disease may also impact the success rates of mobile health apps. Future studies should employ more consistent methods and incorporate diverse patient groups to understand app performance characteristics accurately. The study found a strong correlation between reminders, health monitoring, AI interventions, and adherence; however, more research is required to determine the optimal channel. Future studies should compare app features to discover what makes these therapies work.

These scoping review findings are reliable due to their rigorous and transparent process. PRISMA-ScR compliance facilitated a comprehensive and repeatable search across multiple databases, thereby enhancing internal validity. Two reviewers screened and extracted data to prevent bias, and the inclusion criteria were clearly defined. The synthesis was reinforced by using only real-world data from randomised controlled trials, observational research, and mixed-methods designs. The diversity of individuals, situations, and intervention types makes outcomes more externally valid and generalizable, even if contextual considerations should be taken into account.

Despite the best efforts, it may have overlooked relevant studies in specialist healthcare periodicals not indexed in the study databases. PubMed, Scopus, Web of Science, and IEEE Xplore were included in our assessment due to their broad coverage, comprehensive indexing of academic works, and substantial weighting in the fields of medicine and technology. PubMed, which focuses on clinical and biomedical research; Scopus and Web of Science, which index social sciences, health informatics, and digital health; and IEEE Xplore, which captures literature on health informatics, digital health, and AI-driven technologies, were included. In addition to the study’s main search, the researchers manually screened reference lists and grey literature, including WHO reports and conference proceedings, to avoid missing crucial material from more specialised sources.

Conclusions and future directions

This investigation shows how mobile health applications improve patient health and medication adherence. In particular, mobile health apps with automated reminders, real-time health monitoring, and AI-powered personalised medicines enhance adherence for patients with chronic illnesses, including diabetes, hypertension, and HIV. The analysis suggests that digital therapies increase drug adherence, which improves disease management and health indices, including QoL, glucose stability, and blood pressure. These findings indicate that mobile health-based solutions may help combat non-adherence, a significant global healthcare issue, in a scalable and cost-effective manner. The study also demonstrates the importance of user interaction in achieving app success. Apps with tailored feedback, interesting features, and seamless integration with wearable devices are more effective at long-term tracking than reminder-only apps. Despite overwhelming evidence that mobile health treatments work, the research identifies various barriers to their wider adoption, including a lack of technical understanding, concerns about personal data privacy, and limited financial resources. These restrictions underscore the need for more inclusive and flexible mobile health solutions that cater to diverse patient populations in low-resource settings where digital healthcare remains limited. Health technology is evolving toward data-driven, AI-powered, and patient-centred applications. Machine learning algorithms that predict adherence identify gaps and provide specific health recommendations to inform personalised healthcare. Integrating electronic health records and direct doctor-patient interaction may increase the effectiveness and durability of mobile health therapy. These technological advances will be crucial for patient participation and treatment effectiveness as digital health continues to grow.

This review identifies digital illiteracy, restricted internet access, data privacy concerns, and scepticism among healthcare providers as common barriers to the adoption of mobile health applications. To overcome these obstacles, context-specific and resource-sensitive strategies are needed. Even in low-resource areas, stripped-down, low-bandwidth programs that can run offline or with a poor connection may be used. Community health workers may educate patients on how to use the app and send them messages to assist them in following their treatment programs. Governments and non-governmental organizations (NGOs) can enhance digital literacy by implementing public health campaigns and locally based training programs for diverse age groups and literacy levels. Local stakeholders, including patients, carers, and frontline healthcare workers, should co-design mHealth solutions to improve trust and usability. This will ensure that the tools are culturally suitable and match their local healthcare system. To protect data privacy, consent forms should be concise and encrypted to ensure confidentiality. Integrating mobile health platforms into health information systems and demonstrating cost-effectiveness via pilot projects may promote scalability and sustainability in underserved areas.

Research and development recommendations

While the findings were promising, further research is necessary to address several outstanding issues. The long-term consequences of mobile health therapies are a key area of research. Future longitudinal studies should investigate how habit-building, app fatigue, and patient demands influence long-term usage habits. Additionally, determining whether patients continue to use mobile health applications beyond initial adoption would be necessary. A comparative investigation is needed to determine the optimum health aspects. Mobile health applications may benefit from a more systematic assessment of AI, gamification, social support, and behavioural nudges. Additionally, research must evaluate hybrid therapies combining mobile health apps with traditional healthcare models, including telemedicine consultations and pharmacist-led prescription reviews.

Demographic and socioeconomic aspects that affect mobile health efficacy should also be studied. Most research excludes disadvantaged groups, including older people, low-income residents, and those with minimal digital literacy, in favour of mobile and internet users. Future research should investigate how healthcare inequities, cultural beliefs, and socioeconomic status influence mobile health usage and effectiveness. Researchers should also develop simpler medication adherence apps to make them more accessible to a wider audience. Regulators and policymakers need further study to set standards and best practices for mobile health app development and integration into healthcare systems. AI-driven health solutions are developing rapidly, necessitating stringent data privacy regulations to safeguard patient data. Due to the increasing volume of sensitive health data collected and analysed by mobile health apps, future research should prioritise cybersecurity improvements. Policymakers want healthcare providers and patients to adopt mobile health apps, particularly in public health campaigns, so they should consider payment laws and subsidisation models.

Most research on mobile health applications for medication adherence has focused on short-term benefits, generally a few weeks to several months. The Lack of longitudinal data makes it challenging to evaluate long-term clinical outcomes, adherence, and behavioural changes across patient groups. The difficulty and cost of conducting long-term cohort or randomised controlled trials, as well as the effects of user fatigue and attrition on app engagement over time. The rate of technological change renders mobile health tools outdated or drastically improved before such evaluations can be finished. The lack of integration between app usage data and electronic health records makes it difficult to measure the health impacts of all understudied areas. Future research should focus on longitudinal and hybrid implementation-effectiveness studies to address these obstacles. Study designs should adapt to app updates. Explore EHR-connected real-world data to track patient trajectories following the initial intervention. Funding methods and regulatory incentives may support long-term investigations of mobile health solutions in real-world healthcare settings.

Ultimately, policymakers, software developers, and healthcare providers must collaborate to innovate digital medication management. Future research should integrate public health, medicine, behavioural science, and AI to develop patient-centred digital health solutions. As mobile health technologies continue to advance, medication adherence may become a digitally optimised, patient-driven success story.

Supplemental Information

Supplemental Information 1 PRISMA Checklist.

Additional Information and Declarations

Competing Interests

The authors declare that they have no competing interests.

Author Contributions

Xuye Wang conceived and designed the experiments, performed the experiments, analyzed the data, prepared figures and/or tables, authored or reviewed drafts of the article, and approved the final draft.

Beibei Wang conceived and designed the experiments, performed the experiments, analyzed the data, authored or reviewed drafts of the article, and approved the final draft.

Wan Yin Tew conceived and designed the experiments, performed the experiments, analyzed the data, authored or reviewed drafts of the article, and approved the final draft.

Xiaoning Yang conceived and designed the experiments, analyzed the data, prepared figures and/or tables, authored or reviewed drafts of the article, and approved the final draft.

Xiangyang Xu conceived and designed the experiments, performed the experiments, prepared figures and/or tables, and approved the final draft.

Yifang Gao performed the experiments, analyzed the data, prepared figures and/or tables, and approved the final draft.

Yongjia Chen conceived and designed the experiments, authored or reviewed drafts of the article, and approved the final draft.

Mun Fei Yam conceived and designed the experiments, authored or reviewed drafts of the article, and approved the final draft.

Data Availability

The following information was supplied regarding data availability:

This is a literature article.

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
