# Peer review of "Exploring mHealth interventions for medication management: a scoping review of digital tools, implementation barriers, and patient outcomes"

_PeerJ Computer Science, doi:10.7717/peerj-cs.3190_

## Round 0.1 · original submission · Major Revisions

The article has some value and merit but the reviewers identified a number of issues that need to be addressed and solved before acceptance. I invite the authors to leverage the comments of the reviewers to prepare an improved, enhanced version of the manuscript.

Reviewer 1 ·

Basic reporting

This manuscript reports a scoping literature review of mobile health (mHealth) applications designed to improve medication adherence, with a focus on AI-driven features and their impact on patient outcomes. The methods follow a PRISMA-ScR approach. The authors conclude that mHealth apps hold considerable promise for chronic disease management, but stress the need for inclusive design, strong data privacy measures, and integration into healthcare systems. They recommend future research on long-term effectiveness, cost-effectiveness, and broader implementation strategies. The manuscript addresses a timely and important topic. However, there are several areas where clarity could be improved.

Major Issues:
1. The current title emphasizes AI and medication adherence, but the manuscript primarily focuses on mHealth interventions, with only limited and superficial coverage of AI-related content and adherence-specific outcomes. Instead, the review covers broader topics such as effectiveness and implementation, which fall outside the scope implied by the title. To align with the actual content, the AI discussion should be substantially expanded, and the title should be revised to more accurately reflect the review’s true focus.
2. The methodology lacks critical details. Eligibility criteria (e.g., language, study design, time frame) are insufficiently defined. The search strategy omits search terms, database limits, and grey literature inclusion. The rationale for only including studies between 2022–2024 is unclear and must be justified.
3. The use of a “panel of experts” for assessing methodological quality is atypical for scoping reviews. It is unclear whether a formal risk-of-bias appraisal was conducted. If studies were excluded based on quality, the criteria and justification should be stated. Otherwise, such evaluation should not be implied.
4. Despite AI being highlighted in the title, its discussion is underdeveloped. The review notes that ~25% of studies used AI/ML but fails to critically examine its role (e.g., validation, biases, explainability) or quantify how many studies rigorously assessed AI-driven outcomes. A deeper synthesis of AI’s challenges (e.g., data requirements, regulatory gaps) may also be helpful.
5. Percentages should always include the numerator and denominator to clarify scope and avoid overgeneralization.
6. Technology acceptance, inclusive design, and patient-centered design are very important aspects. Please expand the discussion to discuss them with citations as appropriate.

Detailed Comments:
7. The keywords “artificial healthcare intelligence” are unusual; consider “artificial intelligence in healthcare.”
8. The Introduction should clearly define what constitutes “AI” in this context (e.g., machine learning, predictive algorithms). This will help align the reader’s expectations with later discussions.
9. In the Introduction, “Medicine adherence” should be “Medication adherence.”
10. The Introduction section is overly long and redundant. Some paragraphs are lengthy.
11. Sometimes the term “mHealth applications” is hyphenated or capitalized differently. Decide on one style (e.g., “mHealth applications” vs. “mobile health applications”) and use it consistently.
12. The term “quality of life” appears both in full and as “QoL” in different sections. If using the abbreviation, define it at first mention and apply it uniformly thereafter.
13. The PRISMA diagram (Figure 1) contains display issues with the label “excluded: Methodological issues.”
14. Some expressions (e.g., “educate yourself about...”) are informal and unsuitable for academic writing. Please review the manuscript for tone and revise to maintain scholarly language throughout.

Experimental design

-

Validity of the findings

-

Reviewer 2 ·

Basic reporting

More background information on the novelty of combining AI and healthcare data could be included in the introduction. It would be beneficial to clarify and describe how this technique strategy differs from other digital health initiatives.

Additional citations could support the arguments made in some sections, such as the discussion of adoption barriers. Adding more references to back up these claims would increase the analysis's depth. The manuscript needs a few minor grammatical corrections to improve readability and flow.

More information about the methods, algorithms, or tools used to resolve conflicts during study selection should be included in the methodology section. It would be beneficial to provide a more thorough explanation of the resolution process rather than just mentioning that a third reviewer was consulted.

It's not clear if the authors evaluated the included studies' quality. Did the authors employ a grading scheme, such as GRADE, to evaluate the quality of the included studies?

Also, address whether there is a chance of overlooking pertinent research from specialized healthcare journals and provide a brief explanation of why they chose particular databases for their search.

How effective are mHealth apps in the long run? Analyzing the reasons behind the limitation of research on long-term effects and offering solutions for future research would be beneficial.

It's good that the manuscript shows adoption barriers; it would also be helpful to go into more detail about how to get past them, particularly in environments with limited resources. This section would be more useful if it included suggestions or workable solutions.

A more thorough comparison of AI-driven interventions with conventional techniques.

Using visual aids, such as pie charts to illustrate obstacles and bar charts to demonstrate improvements in adherence, can enhance clarity and engagement.

Finally, the study may add more information about patient-reported experiences, patient satisfaction with mHealth apps.

Experimental design

-

Validity of the findings

-

---

## Round 0.2 · Minor Revisions

Please address the comments of the first reviewer from the previous revision.

Reviewer 1 ·

Basic reporting

Some of my previous comments remain unaddressed.

The paper contains numerous typos and grammatical errors. It requires professional proofreading.

I can review it again when the comments are addressed.

Experimental design

.

Validity of the findings

.

Reviewer 2 ·

Basic reporting

no comment

Experimental design

no comment

Validity of the findings

no comment

Additional comments

no comment

---

## Round 0.3 · accepted · Accept

The reviewer is satisfied with the recent changes and so I can recommend this article for acceptance.